# Privacy-Preserving Contrastive Explanations with Local Foil Trees

**Thijs Veugen** [1,2,*], **Bart Kamphorst** [1] and **Michiel Marcus** [1]

1 Unit ICT, TNO, 2595 DA The Hague, The Netherlands
2 Services and Cybersecurity Group, University of Twente, 7522 NH Enschede, The Netherlands
* Correspondence: thijs.veugen@tno.nl

**Abstract:** We present the first algorithm that combines privacy-preserving technologies and state-of-the-art explainable AI to enable privacy-friendly explanations of black-box AI models. We provide a secure algorithm for contrastive explanations of black-box machine learning models that securely trains and uses local foil trees. Our work shows that the quality of these explanations can be upheld whilst ensuring the privacy of both the training data and the model itself.

**Keywords:** explainable AI; secure multi-party computation; decision tree; foil tree

## 1. Introduction

The field of explainable AI focuses on improving the interpretability of machine learning model behavior. In recent years, exciting developments have taken place in this area, such as the emergence of the LIME [1] and SHAP [2] algorithms, which have become popular. These algorithms take a data point and its classification according to a trained machine learning model and provide an explanation for the classification by analyzing the importance of each feature for that specific classification. This is interesting for a researcher, but a layman using the AI system is unlikely to understand the reasoning of the machine learning model.

Instead, Van der Waa et al. [3] created an algorithm called *local foil trees* that explains why something was classified as class *A* instead of a different class *B* by providing a set of decision rules that need to apply for that point to be classified as class *B*. This increases understanding of the AI system [4] and can, for instance, be used to infer what can be done to change the classification. This is particularly relevant for decision support systems, for which the AI system should provide advice to the user. An example could be that the AI system advises a user to have lower blood pressure and higher body weight in order to go from high risk of a certain illness to a lower risk.

Our work focuses on creating a secure algorithm that provides the same functionality as the local foil tree algorithm in a setting where the black-box machine learning model needs to remain secret to protect the confidentiality of the machine learning model and the training data. Before we explain why this assumption is realistic, we provide a rough overview of the algorithm and interactions in the local foil tree algorithm.

As shown in Figure 1, the user first submits her data to the machine learning model to retrieve a classification. The user then wants to know why she was classified as class *A* and not as class *B*. To create an explanation for this, the explanation provider trains a decision tree and uses the machine learning model as a black-box subroutine within that process. This decision tree is then used to generate an explanation.

In practice, we often see that it can be very valuable to train machine learning models on personal data: for example, in the medical domain to prevent diseases [5], or to detect possible money laundering [6]. Due to the sensitive nature of personal data, however, it is challenging for organizations to share and combine data. Legal frameworks such as the General Data Protection Regulation (https://gdpr-info.eu, accessed on 20 October

2022) (GDPR) and the Health Insurance Portability and Accountability Act (https://www.govinfo.gov/content/pkg/PLAW-104publ191/pdf/PLAW-104publ191.pdf, accessed on 20 October 2022) (HIPAA) further restrict the usage and exchange of personal data.

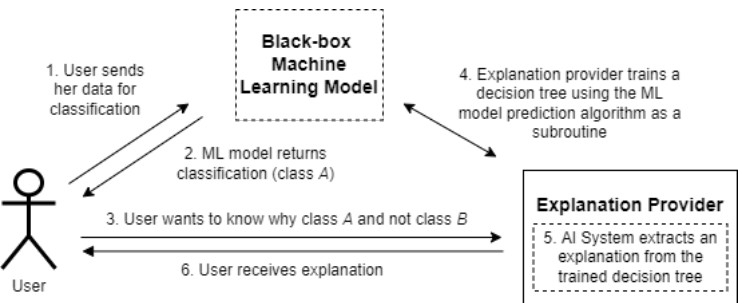

**Figure 1.** Overview of steps and interactions in the local foil tree algorithm.

In order to avoid violating privacy when we want to use personal data as training data for a machine learning algorithm, it is possible to apply cryptographic techniques to securely train the machine learning model, which results in a hidden model [5,7,8]. This ensures that the privacy of the personal data is preserved while it is used to train the model. In order to enable explainable AI with the hidden model, we could simply reveal the model and apply, e.g., the original local foil tree algorithm. However, there are various reasons why it could be undesirable to reveal the trained model. Firstly, if one or more organizations involved have a commercial interest in the machine learning model, the model could be used in ways that were not originally intended. Keeping the model secret then ensures control of model usage. Secondly, sensitive data are used to train the machine learning model, and recent research has shown that it is feasible to reconstruct training data from a trained model [9–11]. The whole reason to securely train the model is to avoid leaking sensitive data, but if the machine learning model is known, it is still possible that sensitive data are leaked when such reconstruction attacks are used. In these cases, we should therefore assume that the model stays hidden to protect the confidentiality of the machine learning model and the training data.

This poses a new challenge for black-box explainable AI. In step 2 of Figure 1, the classification *A* can be revealed to the user without problems, but it is unclear how steps 4 and 5 from Figure 1 would work when the model is hidden. There is a variety of cryptographic techniques that can be used to securely train models. When multiple organizations are involved, common techniques are secret sharing [12] and homomorphic encryption [13]. In this work, we address the aforementioned challenge and provide an algorithm that can produce contrastive explanations when the model is either secret-shared or homomorphically encrypted. Practically, this means that the explanation provider, as shown in Figure 1, does not have the model locally, but that it is owned by a different party or even co-owned by multiple parties. The arrows in the figure then imply that communication needs to happen with the parties that (jointly) own the model.

An additional challenge comes from the fact that explainable AI works best when rule-based explanations, as provided through the local foil tree algorithm, are accompanied by an example-based explanation, such as a data point that is similar to the user but is classified as class *B* instead of *A* [4]. The use of a (class B) data point from the sensitive training data would violate privacy in the worst way possible. As we discuss in Section 3, we address this challenge using synthetic data.

In summary, we present a privacy-preserving solution to explain an AI system, consisting of:

- A cryptographic protocol to securely train a binary decision tree when the target variable is hidden;
- An algorithm to securely generate synthetic data based on numeric sensitive data;

- A cryptographic protocol to extract a rule-based explanation from a hidden foil tree and construct an example data point for it.

The target audience for this work is twofold. One the one hand, our work is relevant for data scientists who want to provide explainable, data-driven insights using sensitive (decentralized) data. It gives access to new sources of data without violating privacy when explainability is essential. On the other hand, our work provides a new tool for cryptographers to improve the interpretability of securely trained machine learning models that have applications in the medical and financial domain.

In the remainder of this introduction, we discuss related work and briefly introduce secure multi-party computation. In the following sections, we explain the local foil tree algorithm [3] and present a secure solution. Thereafter, we discuss the complexity of the proposed solution and share experimental results. Finally, we provide closing remarks in the conclusion.

### 1.1. Related Work

This work is an extended version of [14]. Our solution is based on the local foil tree algorithm by Van der Waa et al. [3], for which we design a privacy-preserving solution based on MPC. There is related work in the area of securely training decision trees, but these results are never applied to challenges in explainable AI. As we elaborate further in Section 3, we have a special setting for which the feature values of the synthetic data to train the decision tree are not encrypted, but the classifications of these data points are encrypted. As far as we know, no training algorithm using such a setting has been proposed yet.

We mention the work of de Hoogh et al. [15], who present a secure variant of the well-known ID3 algorithm (with discrete variables). Their training data points remain hidden, whereas in our case, that is not necessary. Furthermore, as the number of children of an ID3 decision node reflects the number of categories of the chosen feature, the decision tree is not completely hidden. The authors implement their solution using Shamir sharing with VIFF, which is a predecessor of the MPyC [16] framework that we use.

A more recent paper on secure decision trees is by Abspoel et al. [17], who implement C4.5 and CART in the MP-SPDZ framework. We also consider CART since this leads to a binary tree, which does not reveal information on (the number of categories of) the feature chosen in a decision node. Abspoel et al. use both discrete and continuous variables, similar to our settings. However, since Abspoel et al. work with encrypted feature values, they need a lot of secure comparisons to determine the splitting thresholds.

In a similar approach, Adams et al. [18] scale the continuous features to a small domain to avoid the costly secure comparisons at the expense of a potential drop in accuracy.

Only one article was found on privacy-preserving explainable AI. The work of [19] presents a new class of machine learning models that are interpretable and privacy-friendly with respect to the training data. Our work does not introduce new models but provides an algorithm to improve the interpretability of existing complex models that have been securely trained on sensitive data.

### 1.2. Secure Multi-Party Computation

We use secure multi-party computation (MPC) to protect secret data such as the ML classification model and its training data. MPC is a cryptographic tool to extract information from the joint data of multiple parties without needing to share their private data with other parties. Introduced by Yao in 1982 [20], the field has developed quickly, and various platforms are available now for arbitrary secure computations on secret data, such as addition, subtraction, multiplication and comparison. We use the MPyC platform [16] that uses Shamir secret sharing in the semi-honest model, where all parties are curious but are assumed to follow the rules of the protocol.

Like many MPC platforms, MPyC follows the share–compute–reveal paradigm. Each party first uploads its inputs by generating non-revealing shares for the other parties. When the inputs have been uploaded as secrets, the parties can then perform joint computations

without learning the inputs. Finally, the output that is eventually computed is revealed to the entitled parties.

*1.3. Notation*

Due to the inherent complexity of both explainable AI and cryptographic protocols, we require many symbols in our presentation. These symbols are all introduced in the body of this paper; however, for the reader's convenience, we also summarize the most important symbols in Table 1.

**Table 1.** Notation used throughout the document. Some symbols are seen in the context of a certain point (node) within the decision tree, in which case they can be sub- or superscripted with *l* or *r* to denote the same variable in the left or right child node, respectively, that originates from the current node.

| | |
|---|---|
| $A$ | Fact (class); classification of the user as indicated by the black-box. |
| $B$ | Foil (class); target class for contrastive explanation to the user. |
| $\mathcal{B}$ | Decision tree or, equivalently, foil tree. |
| $G_s$ | Gini index for split $s \in \{1, \ldots, \varsigma\}$. |
| $\tilde{G}_s = N_s / D_s$ | Adjusted Gini index for split $s \in \{1, \ldots, \varsigma\}$. |
| $k_A, k_B$ | Index of classes $A$ and $B$, respectively. |
| $K$ | Number of classes. |
| $n$ | Number of available synthetic data points in a particular node. |
| $N$ | Number of synthetic data points $|\mathcal{X}|$. |
| $P$ | Number of features per data point. |
| $\varsigma$ | Number of splits $|\mathcal{S}|$. |
| $S_s = (p_s, t_s)$ | Feature index $p_s \in \{1, \ldots, m\}$ and threshold $t_s$ of split $S_s$, $1 \le s \le \varsigma$. |
| $x_i, x_U$ | Vector $(x_{i,1}, \ldots, x_{i,P})$ of feature values of synthetic data point $i$. With subscript $U$, it refers to the data point of the user. |
| $\mathcal{X}$ | Set of all synthetic data points $x_i$, $i = 1, \ldots, N$. |
| $y_i$ | Indicator vector $(y_{i,1}, \ldots, y_{i,K})$ of the class of data point $i$ as indicated by the black-box. |
| $\xi_i$ | Bit that indicates whether data point $i$ is available (1) or unavailable (0) in the current node. |

Sets are displayed in curly font, e.g., $\mathcal{X}$, and vectors in bold font, e.g., $x_U$. The vector $e_j$ represents the $j$-th elementary vector of appropriate, context-dependent length. The notation $(x \ge y)$ is used to denote the Boolean result of the comparison $x \ge y$. Any symbol between square brackets $[\cdot]$ represents a secret-shared version of that symbol. Finally, a reference to line $y$ of Protocol $x$ is formulated as line $x.y$.

## 2. Explainable AI with Local Foil Trees

In this section, we present the local foil tree method of Van der Waa et al. [3] and discuss the challenges that arise when the black-box classifier does not yield access to its training data and provides classifications in secret-shared (or encrypted) form to the explanation provider.

We assume that we have black-box access to a classification model. If a user-supplied data point $x_U$ is classified as some class $A$, our goal is to construct an explanation why $x_U$ was not classified as a different class $B$. The explanation will contain decision rules in the form that a certain feature of $x_U$ is less (or greater) than a certain threshold value. An overview of the different steps is illustrated in Figure 2 and formalized in Protocol 1. Note that we deviate from Van der Waa et al. by providing an example data point in the final step. In each step of the protocol, we also refer to the section of our work where we present secure protocols for that step.

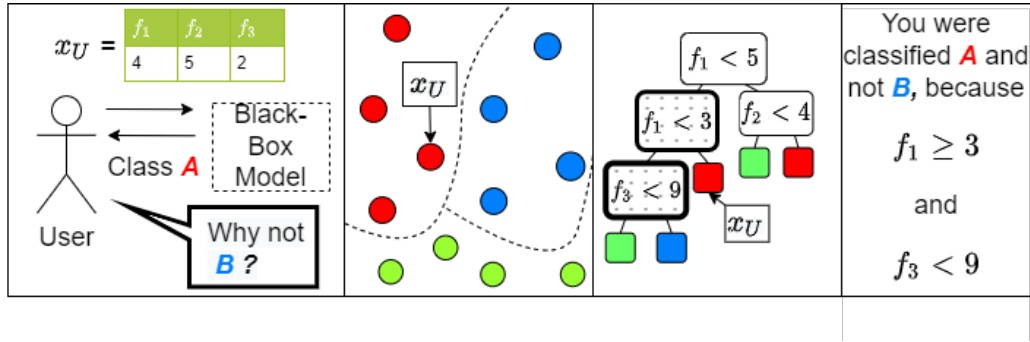

**Figure 2.** A visualisation of the different steps in the local foil tree algorithm to explain why data point $x_U$ was classified as (fact) class $A$ and not as (foil) class $B$. The different images depict classification retrieval and foil class selection, data preparation, decision tree training and determining relevant nodes, and explanation extraction.

---

**Protocol 1** Foil-tree based explanation

---

**Input:** Data point $x_U$ that is classified as class $A$; foil class $B$
**Output:** Explanation why $x_U$ was not classified as the foil class

1: Obtain a classification for the user     ▷ cf. Section 3.1
2: Prepare the synthetic data points for the foil tree     ▷ cf. Section 3.2
3: Classify all synthetic data points through the black-box     ▷ cf. Section 3.3
4: Train a decision tree     ▷ cf. Section 3.4
5: Locate fact leaf (leaf node of $x_U$)     ▷ cf. Section 3.5
6: Determine the foil leaf (leaf node of class $B$ closest to fact leaf)     ▷ cf. Section 3.6
7: Determine the decision node at which the root-leaf paths of the fact and foil leaf split ▷ cf. Section 3.7
8: Construct the explanation (and provide example data point).     ▷ cf. Section 3.7

---

Both limitations that the model owner introduced help to better preserve the privacy of the training data and to secure the model itself, but they also hinder us in generating an explanation. In particular, if we cannot access training data, we need to generate synthetic data that can be used to train a decision tree. Training a decision tree in itself is complicated by the fact that the classifications are hidden, which most notably implies that during the recursive procedure, we need to securely keep track of the synthetic data samples that end up in each branch of the tree.

Information can also be revealed through the structure of the decision tree; in particular, it may disclose the splitting feature. For example, if a certain categorical feature can assume six values and a decision node splits into six new nodes, it is likely that this node represents that feature. For this reason, we do not use the commonly used ID3 or its successor C4.5 for training the decision tree. We instead generate a binary decision tree with the CART (Classification and Regression Trees) algorithm [21]. The CART algorithm greedily picks the best decision rule for a node. In case of classification trees, this materializes as the rule with the lowest Gini index. The Gini index measures the impurity, i.e., the probability of incorrectly classifying an observation, so the lowest Gini index results in the best class purity.

The result of the training procedure is a decision tree whose decision rules and leaf classification are secret-shared. As a consequence we need a secure protocol for determining the position of a foil data point and all nodes that are relevant for the explanation. With help from the model owner(s), among all secret values in the process, only these nodes and the user classification are revealed.

Compared to secure protocols for training decision trees on hidden data points with hidden classification, the fact that we use synthetic data also has some benefits. First, since

the (synthetic) data points are known, we can still access their features directly, improving the efficiency of the protocol. Second, since we already trained the decision tree on synthetic data, we can also supplement our explanation with a synthetic data point, and thereby increase user acceptance [4].

Although our goal is only to provide a privacy-preserving implementation of the local foil tree method, we also mention an evaluation of the method on three benchmark classification tasks [3]. Van der Waa et al. tested the local foil tree method on the Iris dataset, as depicted in Table 2. The column "Mean length" denotes the average length of the explanation in terms of decision nodes, with the total number of features as an upper bound. The accuracy is the $F_1$ score of the foil-tree for its binary classification task on the test set compared to the true labels, and the fidelity score is the $F_1$ of the foil-tree on the test set compared to the model output. For more explanations on this evaluation, we refer to [3].

**Table 2.** Performance of foil-tree explanations of classification tasks on the Iris dataset.

| Classification Model | $F_1$ Score | Mean Length | Accuracy | Fidelity |
|---|---|---|---|---|
| Random forest | 0.93 | 1.94 (4) | 0.96 | 0.97 |
| Logistic regression | 0.93 | 1.50 (4) | 0.89 | 0.96 |
| SVM | 0.93 | 1.37 (4) | 0.89 | 0.92 |
| Neural network | 0.97 | 1.32 (4) | 0.87 | 0.87 |

## 3. Secure Solution

In this section, we describe the secure version of the local foil tree algorithm, which reveals negligible information about the sensitive training data and black-box model. In the rest of this work, we refer to *training data* when we talk about the data used to train the black-box machine learning model and to *synthetic data* when we refer to the synthetically generated data that we use to train the foil tree.

The secure protocol generates $N$ synthetic data points $x_i$, $i = 1, \ldots, N$ with $P$ features that each can be categorical or continuous. To increase the efficiency of the secure solution, we make use of one-hot or integer encoding to represent categorical values. We assume that the class $k \in \{1, \ldots, K\}$ of data point $x_i$ is represented by a secret binary indicator vector $[y_i] = ([y_{i,1}], \ldots, [y_{i,K}])$ such that $y_{i,k} = 1$ if data point $x_i$ is classified as class $k$ by the black-box, and $y_{i,k} = 0$ otherwise.

During the decision training, we maintain an indicator vector $\xi$ of length $N$ such that $\xi_i = 1$ if and only if the $i$-th synthetic data point is still present in this branch.

### 3.1. Classify User Data

We assume that the user is allowed to learn the black-box classification of her own data point $x_U$, so this step is trivial. Without loss of generality, we assume that the user received classification $A$.

### 3.2. Generating Synthetic Data

Van der Waa et al. [3] mention that synthetic data could be used to train the local foil trees, and they suggest using normal distributions. In this section, we apply that suggestion and provide a concrete algorithm for generating a local dataset around the data point for which an explanation is being generated.

We first take a step back and list what requirements the synthetic data should adhere to:

1. The synthetic data should reveal negligible information about the features of the training data.
2. The synthetic data should be local, in the sense that all data points are close to $x_U$, the data point to be explained.
3. The synthetic data should be realistic, such that they can be used in an explanation and still make sense in that context.

State-of-the-art synthetic data generation algorithms such as GAN [22] and SMOTE [23] can generate very realistic data, but they need more than one data point to work, so we cannot apply them to the single data point to be explained. One could devise a secure algorithm for GAN or SMOTE and securely apply it to the sensitive data, but this would affect the efficiency of our solution. In this article, we pursue the simpler approach that was suggested by Van der Waa et al.

Ideally, one would securely calculate some statistics of the sensitive training data for the black-box model and reveal these statistics. Based on these statistics, one could generate a synthetic dataset by sampling from an appropriate distribution. Our implementation securely computes the mean and variance of every feature in the training data and samples synthetic data points from a truncated normal distribution. The reason for truncating is two-fold: first, it allows us to sample close to the user's data point $x_U$, and second, features may not assume values on the entire real line. Using a truncated normal distribution allows us to generate slightly more realistic data that is similar to $x_U$. The details are presented in Protocol 2.

---

**Protocol 2** Synthetic data generation.

---

**Input:** Encrypted black-box training set $[\tilde{\mathcal{X}}]$, integer $N$, target data point $x_U$
**Output:** Synthetic dataset $\mathcal{X}$ with cardinality $N$

1: **for** $p = 1, \ldots, P$ **do**                  ▷ Compute mean and variance of feature $p$.
2:     $[\mu_p] \leftarrow 1/|\tilde{\mathcal{X}}| \sum_{\tilde{x_U} \in \tilde{\mathcal{X}}} \tilde{x}_p$;    $[\sigma_p^2] \leftarrow 1/|\tilde{\mathcal{X}}| \sum_{\tilde{x_U} \in \tilde{\mathcal{X}}} (\tilde{x}_p - [\mu_p])^2$
3: **end for**
4: Reveal $\mu$ and $\sigma^2$
5: $\mathcal{X} \leftarrow \varnothing$
6: **for** $i = 1, \ldots, N$ **do**
7:     **for** $p = 1, \ldots, P$ **do**
8:         **repeat** Draw $x_{i,p}$ from $\mathcal{N}(\mu_p, \sigma_p^2)$
9:         **until** $x_{i,p} \in [x_p - 3\sigma_p, x_p + 3\sigma_p]$
10:     **end for**
11:     $\mathcal{X} \leftarrow \mathcal{X} \cup x_i$
12: **end for**
13: Return $\mathcal{X}$

---

To generate more realistic data, one could also incorporate correlation between features, or go even further and sample from distributions that better represent the training data than a normal distribution.

In our experiments, we noticed that an interval of $[x_p - 3\sigma_p, x_p + 3\sigma_p]$ generally yielded a synthetic dataset that was still close to $x_U$ but also provided a variety of classifications for the data points. A smaller interval (for example of size $2\sigma_p$) often resulted in a dataset for which the distribution of classifications was quite unbalanced. The foil class might then not be present in the foil tree, breaking the algorithm. Larger intervals would result in data points that are not local anymore and would therefore yield a less accurate decision tree.

### 3.3. Classify Synthetic Data

All synthetic training data points $x_i$ can now be classified securely by the model owner(s). This results in secret-shared classification vectors $[y_i]$. The secure computation depends on the model, and it is beyond our scope.

### 3.4. Training a Decision Tree

In this section, we explain the secure CART algorithm that we use to train a secure decision tree, which is formalized in Protocol 3. The inputs to this algorithm are:

1.    $\mathcal{X}$: a set of synthetic data points.

2.  $\mathcal{S}$: a set of splits to use in the algorithm. Each split $S_s \in \mathcal{S}, s = 1, \ldots, \varsigma$ is characterized by a pair $(p_s, t_s)$ that indicates that the feature with index $p_s$ is at least $t_s$.
3.  $\tau$: the convergence fraction used in the stopping criterion.
4.  $[\boldsymbol{\xi}]$: a secret binary availability vector of size $N$. Here, $\xi_i$ equals 1 if the $i$-th synthetic data point is available and 0 otherwise.

---

**Protocol 3** `cart`—Secure CART training of a binary decision tree.

---

**Input:** Training set $\mathcal{X}$, split set $\mathcal{S}$, convergence parameter $\tau \in [0, 1]$, secret-shared binary availability vector $[\boldsymbol{\xi}]$
**Output:** Decision tree $\mathcal{B}$

1: $\mathcal{B} \leftarrow \varnothing$
2: $N \leftarrow |\mathcal{X}|$
3: **while** $\mathcal{B}$ is not fully constructed **do**
4:    **for** $k = 1, \ldots, K$ **do**
5:        $[n_k] \leftarrow \sum_{i=1}^{N} [y_{i,k}] \cdot [\xi_i]$             ▷ nr available data points per class
6:    **end for**
7:    $[n] \leftarrow \sum_{k=1}^{K} [n_k]$                ▷ nr available data points
8:    $[n_{k*}] \leftarrow \mathrm{max}(([n_1], \ldots, [n_K]))$
9:    $[e_{k*}] \leftarrow \mathrm{find}([n_{k*}], ([n_1], \ldots, [n_K]))$     ▷ indicates most common class
10:   **if** $[(n \leq \tau \cdot N)]$ or $[(n = n_{k*})]$ **then**     ▷ branch fully constructed
11:       Extend $\mathcal{B}$ with leaf node with class indicator $[e_{k*}]$
12:   **else**                               ▷ branch splits
13:       **for** $s = 1, \ldots, \varsigma$ **do**
14:           $[G_s] \leftarrow \mathrm{adjusted\_gini}(S_s)$
15:       **end for**
16:       $[G_{s*}] \leftarrow \mathrm{max}([\boldsymbol{G}])$
17:       $[e_{k*}] \leftarrow \mathrm{find}([G_{s*}], [\boldsymbol{G}])$              ▷ indicates best split
18:       $[p_{s*}] \leftarrow \sum_{s=1}^{\varsigma} [e_{s*,s}] \cdot p_s$          ▷ feature of optimal split
19:       $[t_{s*}] \leftarrow \sum_{s=1}^{\varsigma} [e_{s*,s}] \cdot t_s$         ▷ threshold of optimal split
20:       $b \leftarrow$ decision node that corresponds with split $([p_{s*}], [t_{s*}])$
21:       $\boldsymbol{\xi}^l \leftarrow \mathrm{left\_child\_availability}(\mathcal{X}, [\boldsymbol{xi}], [p_*], [t_{s*}])$
22:       $[\boldsymbol{\xi}^r] \leftarrow [\boldsymbol{\xi}] - [\boldsymbol{\xi}^l]$
23:       Extend $b$ to the left with result of $\mathrm{cart}(\mathcal{X}, \mathcal{S}, \tau, [\boldsymbol{\xi}^l])$
24:       Extend $b$ to the right with the result of $\mathrm{cart}(\mathcal{X}, \mathcal{S}, \tau, [\boldsymbol{\xi}^r])$
25:       Extend $\mathcal{B}$ with $b$
26:   **end if**
27: **end while**
28: Return $\mathcal{B}$

---

We start with an empty tree and all training data points marked as available. First, the stopping criterion uses the number of elements of the most common class (line 3.8) and the total number of elements in the availability vector (line 3.7). The stopping criterion from line 3.10 is securely computed by $1 - (1 - [(n \leq \tau \cdot N)] \cdot (1 - [(n = n_{k*})])$ and consequently revealed.

If the stopping condition is met, i.e., equal to one, a leaf node with the secret-shared indicator vector of the most common class is generated. In order to facilitate the efficient extraction of a foil data point as mentioned at the start of Section 3, we also store the availability vector $\boldsymbol{\xi}$ in this leaf node. How this indicator vector is used to securely generate a foil data point is discussed in Section 3.8.

If the stopping criterion is not met, a decision node is created by computing the best split (lines 3.13–19) using the adjusted Gini indices of each split in $\mathcal{S}$. We elaborate on computing the adjusted Gini index (lines 3.13–15) later on in this section.

After determining the optimal split, an availability vector is constructed for each child based on this split in lines 3.21–22. For the left child, this is done using the availability

vector $[\xi]$, the indicator vector indicating the feature $p_{s*}$ of the best split $e_{p_{s*}}$ and the threshold $t_{s*}$ of the best split, as explained in Protocol 4. The resulting availability vector $[\xi^l]$ has a $[1]$ in index $i$ if $x_{i,p_{s*}} \leq t_{s*}$. The entry-wise difference with $[\xi]$ then gives the availability vector for the right child. The CART algorithm is then called recursively with the new availability vectors to generate the children of the decision node.

In Protocol 3, we use two yet-unexplained subroutines, namely `max` and `find`. The `max` subroutine securely computes the maximum value in a list using secure comparisons. Thereafter, the `find` subroutine finds the location of the maximum computed by `max` in the list that was input to `max`, which is returned as a secret-shared indicator vector indicating this location. The functions `max` and `find` are already implemented in MPyC. However, since we always use the two in junction, we implemented a slight variation that is presented in Appendix A.

---

**Protocol 4** `left_child_availability`—Indicate the data points that flow into the left child.

---

**Input:** Synthetic dataset $\mathcal{X}$, availability vector $[\xi]$ for the current node, feature indicator vector $[e_{p_s}]$, threshold $[t_s]$
**Output:** Availability vector $[\xi^l]$ for the left child
1:  **for** $i = 1, \ldots, N$ **do**
2:  　　$[x_{i,p_s}] \leftarrow \sum_{p=1}^{P} [e_{p_s,p}] \cdot x_{i,p}$
3:  　　$[\delta_i] \leftarrow [(x_{i,p_s} \leq t_s)]$
4:  　　$[\xi_i^l] \leftarrow [\xi_i] \cdot [\delta_i]$
5:  **end for**
6:  Return $[\xi^l]$

---

3.4.1. Compute the Gini Index for Each Possible Split

We aim to find the split $S^* := S_{s*} = (p_{s*}, t_{s*})$ with the highest class purity, which is equivalent to the lowest Gini index $G_s$. As such, we first need to compute the Gini index for all splits. The Gini index of a split is the weighted sum of the Gini value of the two sets that are induced by the split,

$$G_s = g_s^l \cdot \frac{n^{s,l}}{n} + g_s^r \cdot \frac{n^{s,r}}{n}. \tag{1}$$

Here, $n$ is again the number of available data points in the current node, $n^{s,l}$ is the number of available data points in the left set that is induced by split $S_s$, and $g_s^l$ is the Gini value of the left set that is induced by split $S_s$,

$$g_s^l := 1 - \sum_{k=1}^{K} \left( \frac{n_k^{s,l}}{n^{s,l}} \right)^2, \tag{2}$$

where $n_k^l$ denotes the number of available data points in the left node with class $k$. The symbols $n^{s,l}$, $n_k^{s,l}$ and $g_s^r$ are defined analogously for the right set. For notation convenience, justified as the upcoming derivations concern a fixed index $s$, we drop the superscripts $s$ from the symbol $n$.

We now derive a more convenient expression for the Gini index. Substituting expression (2) into (1) and rewriting yields

$$G_s = \frac{n^l + n^r}{n} - \frac{n^r \sum_{k=1}^{K} (n_k^l)^2 + n^l \sum_{k=1}^{K} (n_k^r)^2}{n \cdot n^l \cdot n^r}. \tag{3}$$

Now, since $n = n^l + n^r$ is independent of the split, minimizing the Gini index over all possible splits is equivalent to *maximizing* the *adjusted Gini index* $\tilde{G}_s$,

$$\tilde{G}_s = \frac{n^r \sum_{k=1}^{K}(n_k^l)^2 + n^l \sum_{k=1}^{K}(n_k^r)^2}{n^l \cdot n^r} =: \frac{N_s}{D_s}. \tag{4}$$

We represent $\tilde{G}_s$ as a rational number to avoid an expensive secure (integer) division. Both the numerator $N_s$ and the denominator $D_s$ are non-zero if the split $S_s$ separates the available data points, e.g., the split induces at least one available data point in each set. Otherwise, either $n^l = 0$ or $n^r = 0$, and $N_s = D_s = 0$, such that $\tilde{G}_s$ is not properly defined. In line 3.16, one could naively let max evaluate $(\tilde{G}_1 < \tilde{G}_2)$ by computing $(N_1 D_2 < N_2 D_1)$. However, this may yield an undesired outcome if one of the denominators equals zero. Appendix B presents two possible modifications that handle this situation.

Protocol 5 shows how the adjusted Gini index can be computed securely. Observe that $[n]$ and $[n_k]$ were already computed for the CART stopping criterion, so they come for free. The computation $[n_k^l]$ can be implemented efficiently as a secure inner product. The computations of $n$, $n^l$, $n^r$ and $n_k^r$ do not require any additional communication. Because the total number of possible spits $N \cdot P$ is much larger than the number $N$ of data points, it makes sense to precompute $[y_{i,k}] \cdot [\xi_i]$ for each $i$ and $k$ such that the computation of $n_k^l$ for each split requires no additional communication.

---

**Protocol 5** `adjusted_gini`—Compute the adjusted Gini index of a split.

---

**Input:** Synthetic dataset $\mathcal{X}$, vector of available transactions $\xi$, split $(p_s, t_s) = S_s \in \mathcal{S}$
**Output:** Encrypted numerator and denominator of adjusted Gini index
$[\tilde{G}_s] = [N_s]/[D_s]$

1: **for** $i = 1, \ldots, N$ **do**
2: $\quad \delta_i \leftarrow (x_{i,p_s} \leq t_s)$ $\qquad\qquad\qquad$ ▷ 1 if data point meets split criterion, else 0
3: **end for**
4: $[n] \leftarrow \sum_{i=1}^{N}[\xi_i], \quad [n^l] \leftarrow \sum_{i=1}^{N}\delta_i \cdot [\xi_i], \quad [n^r] \leftarrow [n] - [n^l]$
5: $[n_k] \leftarrow \sum_{i=1}^{N}[y_{i,k}] \cdot [\xi_i], \quad [n_k^l] \leftarrow \sum_{i=1}^{N}\delta_i \cdot [y_{i,k}] \cdot [\xi_i], \quad [n_k^r] \leftarrow [n_k] - [n_k^l]$
6: Return $[N_s] \leftarrow [n^r] \sum_{k=1}^{K}([n_k^l])^2 + [n^l] \sum_{k=1}^{K}([n_k^r])^2$ and $[D_s] \leftarrow [n^l] \cdot [n^r]$

---

3.4.2. Convergence

In theory, it is possible that at some point during training, the CART algorithm has not yet met the stopping criterion and has no splits available that actually separate the set of available data points. In this case, the algorithm keeps adding useless decision nodes and does not make any actual progress. To prevent ending up in this situation, we can detect it by revealing $(D_{s*} = 0)$ and can take appropriate action. Additionally, a maximum number of nodes or a maximum depth can be set.

*3.5. Locate the Fact Leaf*

Once the decision tree has been constructed, we need to find the leaf that contains the fact $x_U$. As the fact leaf is revealed, the path from the root to the fact leaf is revealed as well. Therefore, we can traverse the decision tree from the root downwards and reveal each node decision. The decision for data point $x_U$ at a given node can be computed similarly to Protocol 4. First, the feature value that is relevant for the current decision node is determined through $[x_{U,p_{s*}}] = \sum_{p=1}^{P}[e_{p_{s*},p}] \cdot [x_{U,p}]$. Second, the secure comparison $[(x_{U,p_{s*}} \leq t_{s*})]$ is performed and revealed. The result directly indicates the next decision node that needs to be evaluated. This process is repeated until a leaf is encountered: the fact leaf.

### 3.6. Locate the Foil Leaf

Since we know the fact leaf and the structure of the decision tree, we can create an ordered list of all tree leaves, starting with the closest leaf and ending with the farthest leaf. We can traverse this list and find the first leaf that is classified as class $B$ without revealing the classes but only whether they equal $B$, i.e., by revealing the Boolean $[(e_{k^*,k_B} = 1)]$ for every leaf. This does not require any extra computations, as these vectors have already been computed and stored during the training algorithm. We use the number of steps between nodes within the decision tree as our distance metric, but as Van der Waa et al. [3] note, there are more advanced options.

### 3.7. Construct the Explanation

Once the fact leaf and the foil leaf have been determined, the lowest common node can be found without any secure computations since the structure of the decision tree is known. We traverse the decision tree from this lowest common node to the foil leaf and reveal the feature and threshold for each of the nodes on that path (the nodes with a thick border and dotted background in Figure 2). For each rule, we determine whether it applies to $x_U$. For instance, if a rule says that $x_{U,i} \geq 3$ and indeed $x_U$ satisfies this rule, then it is not relevant for the explanation.

After this filter is applied, we combine the remaining rules where applicable. For example, if one rule requires $x_{U,i} \geq 3$ and another rule requires $x_{U,i} \geq 4$, we take the strictest rule, which in this case is $x_{U,i} \geq 4$.

### 3.8. Retrieving a Foil Data Point

Finally, we wish to complement the explanation by presenting the user with a synthetic data point that is similar to the user's data point $x_U$ but that is (correctly) classified as a foil by the foil tree. We refer to such a data point as a *foil data point*. Note that it is possible for samples in a foil leaf to have a classification different from $B$, so care needs to be taken in determining the foil sample.

As mentioned in Section 3.4, we assume that for each leaf node we saved the secret-shared availability vector $\xi$ that indicates which data points are present in the leaf node. In Section 3.6, we determined the foil leaf, so we can retrieve the corresponding binary availability vector $\xi^{foil}$. Recall that the $i$-th entry in this vector equals 1 if data point $x_i$ is present in the foil leaf and 0 otherwise. All foil data points $x_{i^*}$ therefore satisfy $\xi_{i^*} = 1$ and are classified as $B$ by the foil tree.

A protocol for retrieving a foil data point is presented in Protocol 6. It conceptually works as follows. First, it constructs a indicator vector for the position of the foil data point. This vector is constructed in an element-wise fashion with a helper variable $\varepsilon$ that indicates whether we already found a foil data point earlier. Second, the secure indicator vector is used to construct the foil data point $s$, which is then revealed to the user.

---

**Protocol 6** `retrieve_foil`—Retrieve foil data point.

---

**Input:** Availability vector $[\xi]$ of the foil leaf, class index $k_B$
**Output:** Foil data point $s$
1: $[\varepsilon] \leftarrow [0]$                                                 ▷ flips to [1] when a foil data point is found
2: **for** $i = 1, \ldots, n$ **do**
3:      $[\delta_i] \leftarrow (1 - [\varepsilon]) \cdot [\xi_i] \cdot [y_{i,k_B}]$
4:      $[\varepsilon] \leftarrow [\varepsilon] + [\delta_i]$
5: **end for**
6: **for** $p = 1, \ldots, P$ **do**
7:      $[s_p] \leftarrow \sum_{i=1}^{N} [\delta_i] \cdot [x_{i,p}]$
8: **end for**
9: Reveal $s$ to the user

---

It is important that the foil data point is only revealed to the user and not to the computing parties since the foil data point can leak information on the classifications of the synthetic data points according to the secret-shared model, which are the values we are trying to protect. In practice, this means that all computing parties send their shares of the feature values in vector $s$ to the user, who can then combine them to obtain the revealed values.

## 4. Security

We use the MPyC platform [16], which is known to be passively secure. The computing parties jointly form the Explanation Provider (see Figure 1) that securely computes an explanation, which is revealed to the user, who is typically not one of the computing parties. The machine learning model is out of scope; we simply assume secret classifications of synthetic data points are available as secret-sharings of the Explanation Provider without any party learning the classifications.

During the protocol, the Explanation Provider learns data point $x_U$ of the user, its class $A$, and the foil class $B$, together with the average and variance of each feature used to generate synthetic dataset $\mathcal{X}$. Furthermore, the (binary) structure of the decision tree, including the fact leaf, foil leaf, and therefore also the lowest common node, are revealed. Other than this, no training data or model information is known by the Explanation Provider.

The explanation—consisting of the feature index and threshold for each node on the path from the lowest common node to the fact or foil leaf—and the foil data point $s$ are revealed only to the user.

## 5. Complexity

For generation of the binary decision tree, the number $\varsigma \approx N \cdot P$ of all possible splits is large and determines the runtime. For each node, we need to compute the Gini index for all $\varsigma$ possibilities and identify the maximum. If we can compute secure inner products at the cost of one secure multiplication, as in MPyC, the node complexity is linear in $\varsigma$ and $K$ and more or less equal to the costs of $\varsigma$ secure comparisons per node. A secure comparison is roughly linear in the number of input bits, which in our case is $\mathcal{O}(\log_2(NK))$.

However, we can always precompute $[y_{i,k}] \cdot [\xi_i]$ for all $i \in \{1, \dots, N\}$ and $k \in \{1, \dots, K\}$, such that the node complexity is linear in $N$ and $K$. The $\varsigma$ secure comparisons per node cannot be avoided though.

The number of nodes of the decision tree varies between 1 (no split) and $\frac{2N-1}{\tau \cdot N}$ (full binary tree). Therefore, the total computational (and communication) complexity is $\mathcal{O}(N^2 \cdot K)$. Although the aim is to obtain a tree of depth $\log_2 N$, the depth $d$ of the tree will vary between 1 (no split) and $\frac{N-1}{\tau \cdot N}$ (only extremely unbalanced splits). At each tree level, we can find the best splits in parallel, such that the number of communication rounds is limited to $\mathcal{O}(N \cdot \log_2 \varsigma)$ (assuming a constant round secure comparison).

Given the decision tree, completing the explanation is less complex and costs at most $d$ secure comparisons.

## 6. Experiments

We implemented our secure foil tree algorithm in the MPyC framework [16], which was also used for most earlier work on privacy-preserving machine learning [5,8,15,24]. This framework functions as a compiler to easily implement protocols that use Shamir secret sharing. It has efficient protocols for scalar–vector multiplications and inner products. In our experiments, we ran MPyC with three parties and used secure fixed point numbers with a 64-bit integer part and 32-bit fractional part. For the secret-shared black-box model, we secret-shared a neural network with three hidden layers of size 10 each. We used the iris dataset [25] as our training data for the neural network (using integer encoding for the target variable) and generated three synthetic datasets based on the iris dataset of sizes 50, 100 and 150, respectively.

Table 3 shows the results of our performance tests. We report the timing in seconds of our secure foil tree training algorithm under 'Tree Training', for explanation construction under 'Explanation', and for extraction of the data point under 'Data Point'. For each of these, we report the average, minimum and maximum times that we observed. The column 'Accuracy' denotes the accuracy of the foil tree with respect to the neural network. This accuracy is computed as the number of samples from the synthetic dataset for which the classifications according to the neural network and the foil tree are equal, divided by the total number of samples ($N$). We do not provide any performance results on the training algorithm or classification algorithm of the secret-shared black-box model (in this case, the neural network), as the performance of the model highly depends on which model is used, and our solution is model-agnostic.

**Table 3.** Performance results (timing in seconds) of our algorithms in MPyC.

| $N$ | Tree Training | | | Explanation | | | Data Point | | | Accuracy |
|---|---|---|---|---|---|---|---|---|---|---|
| | avg | min | max | avg | min | max | avg | min | max | |
| 50 | 20.396 | 19.594 | 21.158 | 0.033 | 0.027 | 0.041 | 0.157 | 0.112 | 0.219 | 0.96 |
| 100 | 94.455 | 93.133 | 95.234 | 0.061 | 0.058 | 0.062 | 0.277 | 0.269 | 0.361 | 0.89 |
| 150 | 130.575 | 129.681 | 131.327 | 0.050 | 0.038 | 0.052 | 0.404 | 0.387 | 0.425 | 0.91 |

We see that the accuracy does not necessarily increase when we use more samples. A synthetic dataset size of 50 seems to suffice for the Iris dataset and shows performance numbers of less than half a minute for the entire algorithm.

## 7. Conclusions

We present the first cryptographic protocol that is able to explain, in a privacy-preserving way, black-box AI models that are trained by sensitive data. The explanation is constructed by means of local foil trees. After generating synthetic data similar to fact data points, a binary tree is securely computed to find the so-called fact and foil leaves. Using both fact and foil leaves, an explanation of the AI model is constructed that explains to the user why she was classified as the fact class and not as the foil class. We additionally provide a synthetic data point from the foil leaf to strengthen the explanation.

Our solution hides the classification model and its training data in order to provide explanations for users without leaking sensitive commercial or private data. We implemented our solution with MPyC on the Iris dataset with different sizes of synthetic datasets. With 50 samples, we achieved an accuracy of 0.96 within half a minute.

**Author Contributions:** T.V. designed the secure solution, B.K. and M.M. implemented and tested it, and together they wrote the paper. All authors have read and agreed to the published version of the manuscript.

**Funding:** This research received no external funding.

**Data Availability Statement:** We used the Iris data set [25] for our experiments.

**Acknowledgments:** The research of this paper has been done within the FATE project, which is part of the TNO Appl.AI program (internal AI program). We additionally thank Jasper van der Waa for his helpful comments and suggestions.

**Conflicts of Interest:** The authors declare no conflict of interest.

## Appendix A. Indicator Vector of Maximum

Given a list of hidden elements $[z] = ([z_1], \ldots, [z_L])$ and a relation $<$ that induces total ordering on its elements, we need to find the maximum $[z_{max}]$ and the indicator vector of the maximum. One way to do this is to securely deduce $[z_{max}]$ and then apply an independent protocol for finding the position of $[z_{max}]$ and returning the result as an indicator vector. This is supported out-of-the-box in several frameworks.

Instead, we suggest storing some artifacts of the first protocol and leveraging them in the second protocol. This is achieved through Protocols A1 and A2. First, `max([z], 1, L)` performs a binary search to compute the maximum through $L - 1$ secure comparisons, the comparison results of which are stored in $[\gamma_s]$, $1 \leq s < L$, followed by `indicator([1], 1, L)` to compute the indicators $[\delta_s]$, $1 \leq s \leq L$ of the maximum.

This approach with logarithmic round complexity is similar to Protocol 5.1 of de Hoogh [24] and are due to Toft [26]. Since both the recursive calls in line A1.5 can be performed in parallel, the number of iterations is reduced from $L$ to $\log_2 L$.

---

**Protocol A1** `max`—Computes the maximum.

> **Input:** Vector $[z]$, indices $s_l$ and $s_r$
> **Output:** Maximum $[\max\{z_s \mid s_l \leq s \leq s_r\}]$, storing comparison results $[\gamma_s]$, $s_l \leq s < s_r$

1: **if** $s_l = s_r$ **then**
2:     $[z_{\max}] \leftarrow [z_{s_l}]$
3: **else**
4:     $\bar{s} \leftarrow (s_l + s_r) \div 2$                             ▷ split at $\bar{s}$, $s_l \leq \bar{s} < s_r$
5:     $[z_l] \leftarrow \mathtt{max}([z], s_l, \bar{s}); \quad [z_r] \leftarrow \mathtt{max}([z], \bar{s} + 1, s_r)$
6:     $[\gamma_{\bar{s}}] \leftarrow [(z_l < z_r)]$                          ▷ Is $z_r$ the largest?
7:     $[z_{\max}] \leftarrow [z_l] + [\gamma_{\bar{s}}] \cdot ([z_r] - [z_l])$
8: **end if**
9: Return $[z_{\max}]$

---

**Protocol A2** `indicator`—Secure maximum indicator vector.

> **Input:** Indicator $[\delta]$, indices $s_l$ and $s_r$, and comparison results $[\gamma]$
> **Output:** Indicators $\{[\delta_s] \mid s_l \leq s \leq s_r\}$ of the overall maximum
> **Invariant:** $\delta = (s_l \leq \mathrm{argmax}\{z_s \mid 1 \leq s \leq L\} \leq s_r)$

1: **if** $s_l = s_r$ **then**
2:     Return $[\delta]$
3: **else**
4:     $\bar{s} \leftarrow (s_l + s_r) \div 2$                             ▷ split at $\bar{s}$, $s_l \leq \bar{s} < s_r$
5:     $[\epsilon] \leftarrow [\delta] \cdot [\gamma_{\bar{s}}]$
6:     $[\delta_l] \leftarrow \mathtt{indicator}([\delta] - [\epsilon], s_l, \bar{s}); \quad [\delta_r] \leftarrow \mathtt{indicator}([\epsilon], \bar{s} + 1, s_r)$
7:     Return the elements of $[\delta_l]$ and $[\delta_r]$
8: **end if**

---

## Appendix B. Comparing Adjusted Gini Indices

Given two splits $S_1$ and $S_2$, we wish to compare their adjusted Gini indices $[\tilde{G}_1] = [\tilde{N}_1]/[\tilde{D}_1]$ and $[\tilde{G}_2] = [\tilde{N}_2]/[\tilde{D}_2]$. In particular, we wish to compute $[(\tilde{G}_1 < \tilde{G}_2)]$ with the interpretation that an index with zero-valued denominator is always smaller than the index. If both denominators are zero, the result does not matter.

To avoid complications when either denominator is zero, we change the straightforward integer comparison $N_1 \cdot D_2 < N_2 \cdot D_1$ to

$$N_1 \cdot (N_1 \cdot D_2 - D_1 \cdot N_2) < 1 - D_1. \tag{A1}$$

To see why this is correct, recall that $N_s$ and $D_s$ are both integers, and $D_s = 0$ if and only if $N_s = 0$. Additionally, it follows from Equation (4) that

$$N_s \geq n^r \sum_{k=1}^{K} n_k^l + n^l \sum_{k=1}^{K} n_k^r = 2D_s. \tag{A2}$$

Therefore, the following statements hold:

- If $D_1, D_2 > 0$, then $N_1, N_2 > 0$ and (A1) evaluates to the result of $N_1 \cdot D_2 - D_1 \cdot N_2 < \frac{1 - D_1}{N_1} \in (-1/2, 0]$. Since all variables on the left-hand side are integers, this is equivalent to $N_1 \cdot D_2 - D_1 \cdot N_2 < 0$.
- If $D_1 > 0, D_2 = 0$, then $N_2 = 0$, and (A1) evaluates to the result of $0 < 1 - D_1$, which is False.
- If $D_1 = 0$, then $N_1 = 0$, and (A1) evaluates to the result of $0 < 1$, which is True.

An alternative approach is to compute $N_1 \cdot (D_2 + 1) < N_2 \cdot (D_1 + 1)$. Theoretically, this comparison might not indicate the worst adjusted Gini index if the indices have a very small difference, but a significant efficiency boost can be expected as the secure comparison input can be represented in fewer bits.

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
