# Peer review of "Privacy-Preserving Contrastive Explanations with Local Foil Trees"

_cryptography, doi:10.3390/cryptography6040054_

Round 1

Reviewer 1 Report

The paper is quite interesting and is well organised, the various points are clearly expressed and the language is correct and understandable. It can be published in present form.

Author Response

Thank you for appreciating our results.

Reviewer 2 Report

The authors presented a combination of privacy-preserving technologies and XAI to enable privacy-inclusive XAI for black box AI models. The research is interesting but I have a few points mentioned below: 

1. They talked about other XAIs such as LIME and SHAP in related works as well as other privacy-preserving solutions.  But a rationale is needed as to why local foil trees, or why Shamir secret sharing instead of Blakeley's or additive sharing is needed. the rationale can be done by providing an extensive comparison between those works and their contribution. 
2.  Evaluation metrics and results are expected and are somewhat absent. 
3. XAI visualization is expected. currently, the paper has algorithms and theories about what it might look like. But an example-wise evaluation is needed on iris data. 

Author Response

Thank you for your valuable comments. Let me explain how we processed them point by point:

  1. The rational of using local foil trees is described in the first two paragraphs of the introduction. Basically, it provides a more user-friendly explanation than LIME and SHAP. The reason that we use Shamir secret sharing is that most prior work on privacy-preserving machine learning has been done with MPyC, and it offers an easy way of implementing the secure solution. In the beginning of Section 6 we added this argumentation.
  2. Our goal is to show how local foil trees could be implemented in a privacy-friendly way. The local foil tree method has been evaluated already in the original paper. We added a summary of this evaluation at the end of section 2.
  3. The XAI method is visualised in Figure 2. The evaluation with iris data has been added at the end of section 2.

Round 2

Reviewer 2 Report

I incline to accept the paper. I have no further points.